# SARS-CoV-2 nsp3 and nsp4 are minimal constituents of a pore spanning replication organelle

Liv Zimmermann [1], Xiaohan Zhao[1], Jana Makroczyova [1], Moritz Wachsmuth-Melm [1], Vibhu Prasad [2], Zach Hensel [3], Ralf Bartenschlager [2,4,5] & Petr Chlanda [1]✉

Coronavirus replication is associated with the remodeling of cellular membranes, resulting in the formation of double-membrane vesicles (DMVs). A DMV-spanning pore was identified as a putative portal for viral RNA. However, the exact components and the structure of the SARS-CoV-2 DMV pore remain to be determined. Here, we investigate the structure of the DMV pore by in situ cryo-electron tomography combined with subtomogram averaging. We identify non-structural protein (nsp) 3 and 4 as minimal components required for the formation of a DMV-spanning pore, which is dependent on nsp3-4 proteolytic cleavage. In addition, we show that Mac2-Mac3-DPUP-Ubl2 domains are critical for nsp3 oligomerization and crown integrity which influences membrane curvature required for biogenesis of DMVs. Altogether, SARS-CoV-2 nsp3-4 have a dual role by driving the biogenesis of replication organelles and assembly of DMV-spanning pores which we propose here to term replicopores.

Positive-strand RNA viruses hijack and remodel the host cell membranes into replication organelles (ROs) to provide a platform for viral RNA synthesis[1–3]. Severe acute respiratory syndrome coronavirus 2 (SARS-CoV-2) induces the formation of endoplasmic reticulum (ER)-derived double-membrane vesicles (DMVs), which serve as ROs[4–6]. Alongside viral proteins, host cell factors involved in autophagy have also been shown to contribute to the formation of DMVs[7,8]. It is well established that coronavirus replication-transcription complex (RTC) is associated with DMVs to orchestrate viral genome replication and transcription yielding subgenomic messenger RNAs (sgRNAs) using double-stranded (ds) RNA as intermediate[4,6,9]. While dsRNA can be directly visualized inside the DMVs[10] and detected by immunolabelling[4,6] it is not clear whether the replication and transcription of viral RNA take place on the inner (luminal) or outer (cytoplasmic) side of the DMV. DMVs may provide a shielded environment for intermediates of viral RNA during replication, evading

recognition of innate immune sensors present in the host cell and thereby facilitating robust viral genome replication and transcription. A structure spanning the DMV induced by the murine hepatitis coronavirus (MHV) was identified and proposed to serve as a putative pore dedicated to the translocation of the newly synthesized viral genomic RNA (gRNA) and sgRNAs from the DMV lumen into the cytoplasm[1]. However, the exact constituents and minimally required protein components of the pore complex in coronavirus replication organelles are still enigmatic. While the non-structural protein (nsp) 3 is the largest multi-domain protein encoded by the coronavirus genome and presumably the major component of the pore complex that constitutes the crown region of the pore as shown for MHV[1], the architecture of the pore spanning DMVs in SARS-CoV-2 infected cells is unknown. It has been shown that the Middle East respiratory syndrome coronavirus (MERS-CoV), SARS-CoV and SARS-CoV-2 nsp3 and nsp4 are sufficient to induce DMV formation[7,11–15] but it remains to be

[1]Schaller Research Group, Department of Infectious Diseases, Virology, Heidelberg University, 69120 Heidelberg, Germany. [2]Department of Infectious Diseases, Molecular Virology, Heidelberg University, 69120 Heidelberg, Germany. [3]ITQB NOVA, Universidade NOVA de Lisboa, 2780-157 Oeiras, Portugal. [4]Division Virus-Associated Carcinogenesis, German Cancer Research Center (DKFZ), 69120 Heidelberg, Germany. [5]German Center for Infection Research (DZIF), Heidelberg partner site, 69120 Heidelberg, Germany. ✉e-mail: petr.chlanda@bioquant.uni-heidelberg.de

determined whether the expression of nsp3 and nsp4 is sufficient to form a pore. Moreover, the mechanism of how these proteins induce membrane curvature to shape ER into DMVs remains to be elucidated. Here we establish a workflow that allowed us to investigate the structure of the SARS-CoV-2 DMV spanning pore and to determine its minimal components using cryo-electron tomography (cryo-ET) and subtomogram averaging in the native cellular environment under low biosafety conditions. We identify nsp3-4 proteins as minimal components of the SARS-CoV-2 DMV pore, identify domains that are important for crown formation and show that the crown integrity has an influence on DMV curvature.

## Results

### SARS-CoV-2 nsp3 and nsp4 are sufficient to form a pore

To determine the minimally required components assembling a DMV pore, we performed in situ cryo-ET on lamellae prepared by focused ion beam (FIB) milling of transfected cells expressing SARS-CoV-2 nsp3-4. To guide cryo-FIB milling and increase our throughput, a plasmid encoding the fluorescent protein mApple was co-transfected with the nsp3-4 construct. Our previous work showed that co-transfection of different constructs leads to efficient co-expression of proteins in transfected cells[16]. In this study, 80.6 % of transfected cells showed co-expression of mApple and nsp3-4 (Supplementary Fig. 1h–j). This allowed us to perform cryo-correlative light and scanning electron microscopy to selectively mill only transfected cells by cryo-FIB milling (Supplementary Fig. 1). We observed a network of DMVs in the cytoplasm of VeroE6 cells clearly showing pores spanning the DMVs (Fig. 1a, b, Supplementary Movie 1). 3D segmentations revealed that nsp3-4 induced DMVs are interconnected by double membrane sheets containing pores which we termed double membrane connectors (DMCs) (Fig. 1a–c, e, Supplementary Fig. 2a). The presence of pores in membranes of DMV connectors has not been described before. In addition, DMVs were frequently connected to ER cisternae decorated by ribosomes, indicating that the DMV network induced by nsp3-4 expression originates from ER membranes similarly as reported for DMVs induced upon SARS-CoV-2 infection[17] (Supplementary Fig. 2b, c). The mean diameter of DMVs formed by nsp3 and nsp4 was 104 nm (SD = 50, $n$ = 62), thus smaller than the DMV mean diameter of 336 nm (SD = 50, $n$ = 20) measured in infected VeroE6 cells[10] (Supplementary Fig. 3a, b) consistent with previous reports[13,17]. In our previous study on SARS-CoV-2 infected cells, we reported a network of filamentous structures with a diameter of 2.7 nm inside DMVs, which is consistent with the size of dsRNA[10]. The lumen of the nsp3-4 induced DMVs did not contain such a network of filamentous structures. We could occasionally detect filamentous structures with a diameter around 4 nm both inside and outside of nsp3-4 DMVs (Supplementary Fig. 4). Hence, in the absence of viral RNA and other viral proteins, nsp3-4 alone is not able to transport and selectively accumulate cellular RNA in the DMV lumen. The data indicates that viral RNA transcription and replication are not prerequisites for DMV and pore formation but might contribute to the size of the DMVs. Noticeably, vesicle packets formed by homotypic DMV-DMV membrane fusion reported previously during late infection[4,10], were not detected in nsp3-4 induced DMVs. Additionally, we observed that some DMVs contain ribosomes and unknown densities (Supplementary Fig. 5), which are similar to those described as dense granules inside DMVs found in poliovirus infected cells suggesting the involvement of autophagic proteins in DMV biogenesis[18]. Importantly, due to the lack of chemical fixation, the structure of the nsp3-4 pores was better preserved in transfected cells than in SARS-CoV-2 infected cells that had to be fixed[1,10] (Fig. 1c, Supplementary Fig. 3a). Strikingly, both DMVs and DMCs contained clearly discernable pores which consist of a double membrane spanning region and of a crown-like assembly facing the convex side of the DMV (Fig. 1c, f). Subtomogram averaging

revealed a pore structure at a resolution of 20 Å and a symmetry scan revealed sixfold rotational symmetry along the axis normal to the DMV surface (Fig. 1g, h, Supplementary Fig. 7a, b). The crown part of the pore showed a 25-nm-wide crown-like structure with 6 prongs decorating the convex cytosolic side of the DMV. Underneath the crown, a channel of 2-3 nm diameter was detected, which is the direct connection to the luminal side of the DMV (Fig. 1i). The overall architecture of the SARS-CoV-2 nsp3-4 pore shares structural similarity with pores in DMVs resulting from MHV infection[1] (Supplementary Fig. 8d). This demonstrates that the expression of nsp3 and nsp4 is sufficient for the formation of the SARS-CoV-2 DMV pore.

### N-terminal domains of SARS-CoV-2 nsp3 are crucial for DMV biogenesis and pore assembly

To further investigate the structure of the pore and the role of nsp3 in DMV biogenesis, we generated two N-terminal nsp3 truncations by deleting the region comprising (i) the ubiquitin-like domain 1 (Ubl1), hypervariable region (HVR) and macrodomain 1 (Mac1) (construct ΔUbl1-Mac1) or (ii) Ubl1, HVR, Mac1–3, Domain Preceding Ubl2 and papain-like protease (PLpro) (DPUP) and Ubl2 (construct ΔUbl1-Ubl2). In addition, to shed light on the importance of the proteolytic cleavage of nsp3 and nsp4 to the pore formation, we substituted GG for AA at the cleavage site at position 1944-1945 (GG > AA), which renders the polyprotein unable to be cleaved[12] (Fig. 2a, b). Western blot analysis of all nsp3-4 constructs showed bands consistent with the predicted molecular masses. Furthermore, Western blot analysis confirmed proteolytic processing at the nsp3-4 GG cleavage site mediated by PLpro which was abrogated by mutation of the GG to AA (GG > AA), (Fig. 2c, d). Interestingly, the deletion in the ΔUbl1-Ubl2 construct slightly reduced nsp3-4 cleavage, indicating that the Ubl2 domain proximal to PLpro is supporting the function of the protease or contributes to proper exposure of the cleavage site (Fig. 2c, d). Confocal microscopy revealed that nsp3 and nsp4 colocalize in all constructs. While nsp3 and nsp4 signals overlapped in the ΔUbl1-Ubl2 construct, the nsp3 signal was lower than in other constructs, presumably because of impaired folding (due to reduced nsp3-4 cleavage) and only partial exposure of the HA-tag at the N-terminus (Fig. 2e–h). The nsp4 signal was more dispersed in the ΔUbl1-Ubl2 construct than in the other constructs which might indicate the fraction of protein that was not cleaved. Expression of all nsp3-4 constructs led to the formation of clusters containing remodeled membranes in cells as determined by electron microscopy analysis of samples processed by high pressure freezing and freeze substitution (Fig. 2i–l). The size and distributions of the clusters were in agreement with the size of the puncta shown by confocal microscopy and native HEK293T cells did not show such remodeled membranes (Supplementary Fig. 6). Notably, while the removal of Ubl1-Mac1 domains does not abrogate DMV formation, removal of Ubl1-Ubl2 domains leads to membrane remodeling but the formation of distinct DMVs is impaired. Consistent with studies on MERS-CoV[12], abrogation of SARS-CoV-2 nsp3-4 cleavage resulted in the formation of large double-membrane whorl-like structures while DMVs were not detected. The whorl-like structures were not observed in untransfected cells (Supplementary Fig. 6).

We next performed in situ cryo-ET on cryo-FIB milled lamellae of transfected (mApple positive) VeroE6 cells to assess the membrane remodeling and determine whether nsp3-4 truncated at the N-terminus is sufficient for pore formation. Consistent with the data from thin-section EM, all nsp3-4 constructs were able to induce membrane remodeling. The expression of the ΔUbl1-Mac1 construct led to the formation of spherical DMVs that contained a similar number of pores but had a larger DMV radius than spherical DMVs induced by unaltered nsp3-4 (Fig. 3a, b, e, f, j, Supplementary Fig. 8f, Supplementary Movie 2). In contrast, the expression of the ΔUbl1-Ubl2 nsp3-4 construct led to remodeling of membranes into a double-membrane

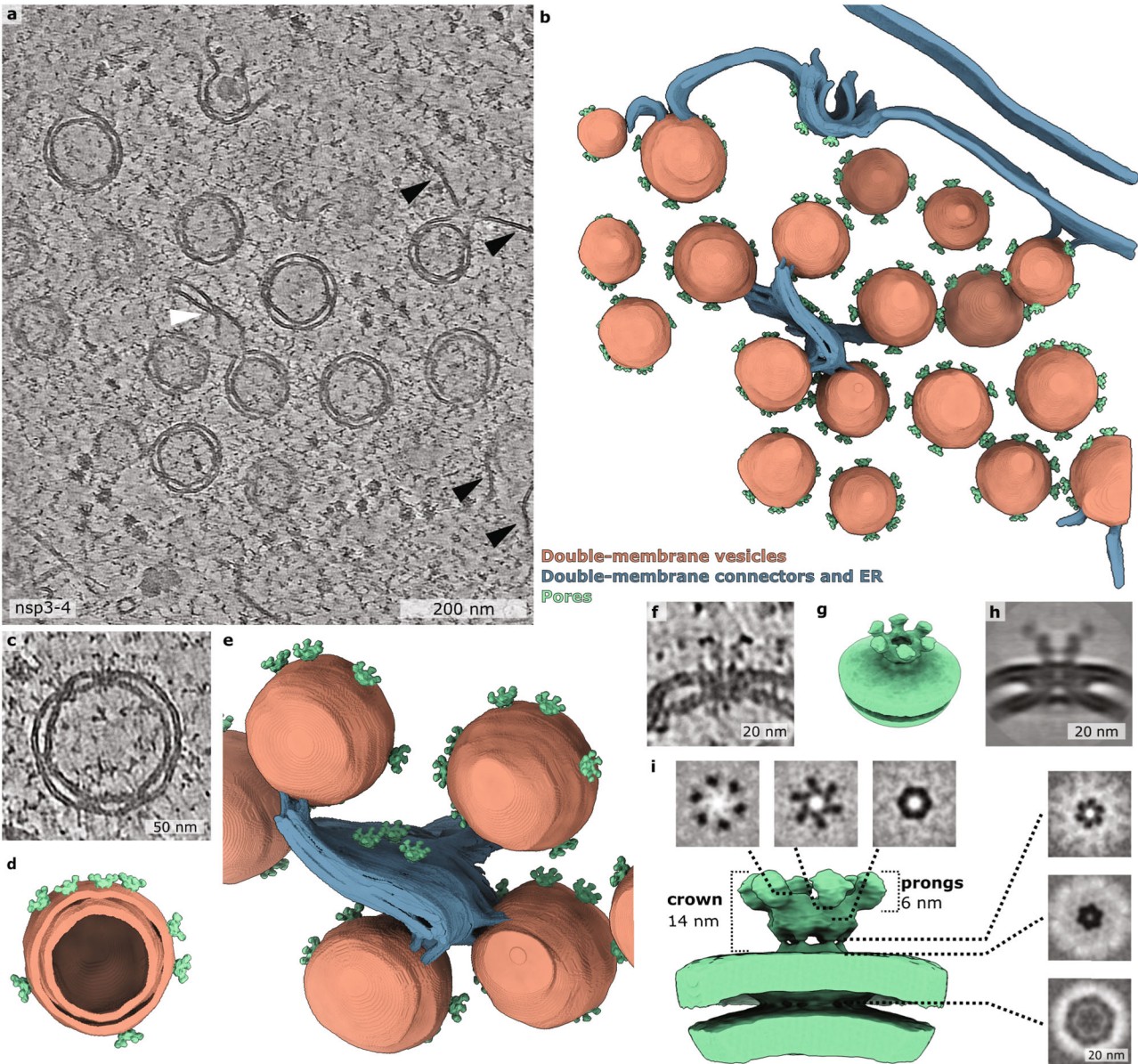

**Double-membrane vesicles**
**Double-membrane connectors and ER**
**Pores**

**Fig. 1 | The SARS-CoV-2 nsp3-4 proteins are sufficient to form DMV-spanning pores. a** Averaged slices of a tomogram acquired on cryo-lamella of VeroE6 cells transfected with HA-nsp3-4-V5 and plunge frozen at 16 hpt. The double-membrane vesicles (DMVs) are interconnected through double-membrane connectors (DMCs) which are highlighted by a white arrow. The connections between DMVs and ribosome-decorated endoplasmic reticulum are highlighted by black arrows. **b** Volume rendering of DMV and DMC network with pores spanning the DMVs and the DMCs. **c** Magnified view of a single DMV with multiple pores spanning two membranes. **d** Volume rendering of the DMV in **c** showing the inside of the DMV. **e** Magnified view of volume rendering showing DMVs interconnected through DMCs containing pores. **f** Averaged slices of a tomogram showing a magnified view of one DMV pore which spans the DMV in **c**. **g** Isosurface of the filtered C6 symmetrized subtomogram average of SARS-CoV-2 HA-nsp3-4-V5 showing the pore complex with its crown on the convex side of the double membrane. **h** Slices averaged through the filtered C6 symmetrized subtomogram average of the pore complex. **i** Slices through the filtered C6 symmetrized subtomogram average from top to bottom. Isosurface of the pore complex induced by SARS-CoV-2 HA-nsp3-4-V5 (green).

network composed of DMCs, which contained pores and formed a mixture of partially closed and completely closed ovoidal structures (Fig. 3c, g, Supplementary Movie 3). Cryo-ET data allowed us to measure DMV radius (r, using the outer membrane), DMV luminal spacing, and calculate the nearest pore-to-pore distance (Fig. 3i–l). In addition, we determined the membrane curvature of the outer DMV membrane (1/r) and counted the number of pores per DMV (Supplementary Fig. 8e–g). DMVs formed by nsp3-4 contained on average approximately 10 (SD = 1.4, n = 31) pores per DMV. This shows that nsp3-4 forms DMVs with a higher density of pores (10 pores per DMV with a diameter of 104 nm) than reported for DMVs formed in MHV infected cells with approximately 8 pores per DMV with a diameter of 257 nm[1].

Noteworthy the number of pores per DMV does not correlate with the DMV diameter (Supplementary Fig. 8g).

## PL^pro cleavage of nsp3-4 is required for pore formation but not for membrane pairing

Interestingly, the double membrane whorl-like structures induced by the cleavage incompetent mutant of nsp3-4 (GG > AA) did not contain any pores (Fig. 3d, h, Supplementary Movie 4) and the luminal spacing was reduced from 16 nm (SD = 1.1, n = 18) to 13 nm (SD = 0.8, n = 19). This indicates that the uncleaved nsp3-4 induces membrane pairing, but pore formation is required for membrane bending and scission which gives rise to DMVs.

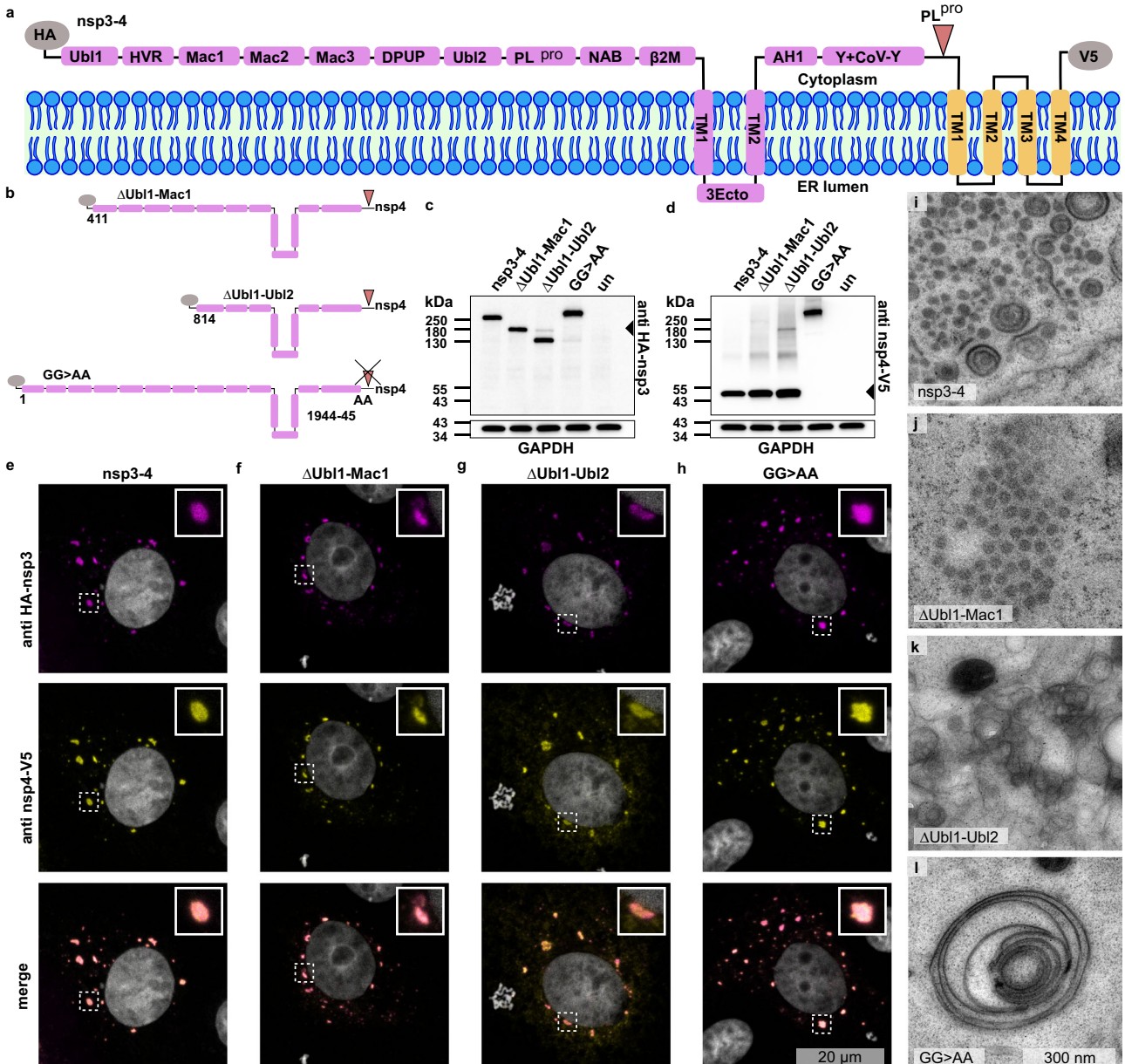

**Fig. 2 | The N-terminal domains of SARS-CoV-2 nsp3 are critical for the formation of distinct DMVs. a** Schematic representation of membrane bilayer with nsp3-4 polyprotein tagged with HA tag at the N-terminus and V5 tag at the C-terminus. Nsp3 domains and nsp4 are shown in magenta and in yellow, respectively. The PL^pro cleavage site between nsp3 and nsp4 polyprotein is indicated by an arrow. **b** Nsp3-4 truncations were generated by deletion of N-terminal domains ΔUbl1-Mac1 and ΔUbl1-Ubl2 and the cleavage mutant GG > AA was generated by substituting GG with AA at the position 1944-1945. All constructs were tagged at the N- and C- termini, with HA and V5 tags, respectively. Western blot analysis, using anti-HA (**c**) and anti-V5 (**d**) staining of lysates of untransfected VeroE6 cells (un) and VeroE6 cells transfected with nsp3-4, ΔUbl1-Mac1, ΔUbl1-Ubl2, or GG > AA construct harvested at 24 hpt. Uncropped blots in Source Data. Representative blots from three biologically independent experiments. Confocal microscopy of VeroE6 cells transfected with nsp3-4 (**e**), ΔUbl1-Mac1 (**f**), ΔUbl1-Ubl2 (**g**) and GG > AA (**h**) chemically fixed at 24 hpt. Representative images of nsp3-4: $n = 17$; ΔUbl1-Mac1: $n = 11$; ΔUbl1-Ubl2: $n = 13$; GG > AA: $n = 10$ cells. Data from one experiment. Nsp3 and nsp4 were detected using anti-HA and anti-V5 antibodies, shown in magenta and yellow, respectively. Boxed images are zoomed areas that are highlighted with dashed boxes. **i–l** Thin-section EM images of HEK293T cells transfected with respective constructs and high-pressure frozen at 24 hpt. Representative images from at least 5 cells showing the phenotype. Data from one experiment. **i** Nsp3-4 induced formation of DMV clusters. **j** Truncation of Ubl1-Mac1 domains does not affect the formation of DMV clusters. **k** Truncation of Ubl1-Ubl2 domains leads to DMV clusters with partially open DMVs. **l** Cleavage mutant shows double-membrane whorl-like structures with wider diameter than DMVs.

## Ubl1-Ubl2 domains of SARS-CoV-2 nsp3 are critical for pore structural integrity

To compare the architecture of DMV pores formed by truncated nsp3-4 constructs, we applied subtomogram averaging of pores detected with ΔUbl1-Mac1 and ΔUbl1-Ubl2 induced DMVs, respectively. Our data showed that while pores formed by nsp3-4 lacking N-terminal Ubl1-Mac1 domains still formed crown-like structures on the convex side of the DMVs, the prongs were partially collapsed and shorter at the distal end in comparison to wildtype nsp3-4 pores, presumably because of altered inter-nsp3 interactions leading to instability of the prongs (Fig. 4a, b, d, e, Supplementary Fig. 8a). Surprisingly, nsp3-4 lacking the Ubl1-Ubl2 domains showed no crown-like structure and only a membrane-proximal density of prongs was visible (Fig. 4c, f, Supplementary Fig. 8b). This data demonstrates that the nsp3 N-terminal

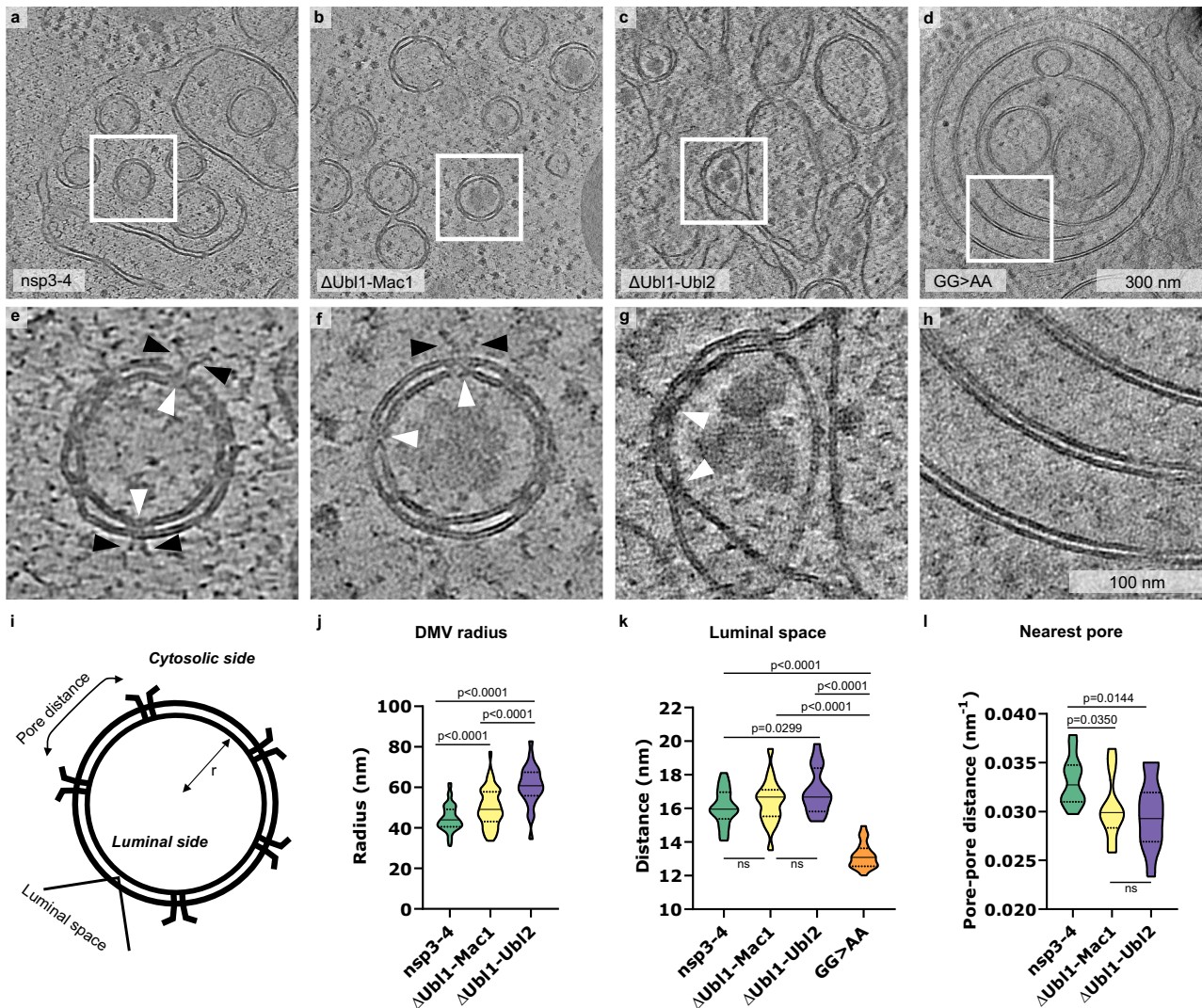

**Fig. 3 | SARS-CoV-2 nsp3 Ubl1-Ubl2 domains and nsp3-4 cleavage site are required for DMV biogenesis.** Slices of tomograms of VeroE6 cells transfected with nsp3-4 (**a**), ΔUbl1-Mac1 (**b**), ΔUbl1-Ubl2 (**c**) and GG > AA (**d**) plunge frozen at 18–24 hpt. **e**–**h** Slices of tomograms displaying magnified areas in (**a**)–(**d**). White arrowheads indicate pores spanning the DMVs, and pore prongs are indicated by black arrowheads. **i** Schematic representation of DMV and indicated parameters that are displayed in plots shown in (**j**)−(**l**). **j** Radius of DMVs in nsp3-4 ($n = 103$), ΔUbl1-Mac1 ($n = 103$), ΔUbl1-Ubl2 ($n = 26$). **k** Luminal spacing between two membranes (nsp3-4: $n = 18$; ΔUbl1-Mac1: $n = 17$; ΔUbl1-Ubl2: $n = 15$; GG > AA: $n = 19$). **l** Nearest pore neighbor distance within one DMV (nsp3-4: $n = 12$; ΔUbl1-Mac1: $n = 12$; ΔUbl1-Ubl2: $n = 11$). Data are shown as a Violin plot indicating the median, 25% and 75% quartiles. Unpaired two-tailed t-test showed significant differences.

region including domains localized between Ubl1 and Ubl2 is critical for the formation of the crown-like structure and pivotal for nsp3-nsp3 interactions, which are likely required for multimerization. Subtomogram averaging highlighted the difference in outer membrane curvature formed by different nsp3-4 constructs. While nsp3-4 was able to induce membrane curvature $\kappa = 0.017$ nm$^{-1}$ of the outer membrane (Fig. 4g), truncations of the nsp3 N-terminus, which are associated with loss of crown structural integrity, caused flattening of the outer membrane. Strikingly, the deletion spanning Ubl1-Ubl2 domains led to a reduction of outer membrane curvature to $\kappa = 0.008$ nm$^{-1}$. In contrast, inner membrane DMV curvature remained unchanged in all constructs (Fig. 4g). To gain further insight in the role of nsp3 N-terminal domains in pore structural integrity, we analyzed the subtomogram averages to determine concave and convex pore membrane curvature. The concave curvature was determined as a reciprocal value of the radius measured in the center of the pore. The measured pore radius was smaller in pores with truncated nsp3 N-terminal domains (r values: nsp3-4: 2 nm, ΔUbl1-Mac1: 1.5 nm, ΔUbl1-Ubl2: 1.1 nm). Therefore, concave pore membrane curvature values

increased upon truncation of N-terminal domains (Fig. 4h). The convex curvature was calculated using imodcurvature tool from the segmented membranes of the pore. While the convex pore membrane curvature was not affected by truncation of Ubl1-Mac1 domains, removing Ubl1-Ubl2 domains led to a decrease of the convex pore curvature (Fig. 4i). Overall, our data show that both DMV and pore membrane curvature is altered in the absence of nsp3 N-terminal domains.

**Model of nsp3 domain localization within the crown of the pore**

The subtomogram average of the crown region allowed us to approximately fit individual nsp3 domains, whose structures were determined experimentally or were predicted using ColabFold[19], following the sequence from N-terminal Ubl1 to betacoronavirus-specific marker (βSM) domain proximal to transmembrane domain 1 (TM1) (Fig. 5a, b). The HVR between Ubl1 and Mac1 domain is not included in the model, as HVR structure predictions have low confidence (pLDDT <50). The subtomogram average-based model of the crown region indicates that Ubl1 and Mac1 are located within the tip of the prong followed by Mac2

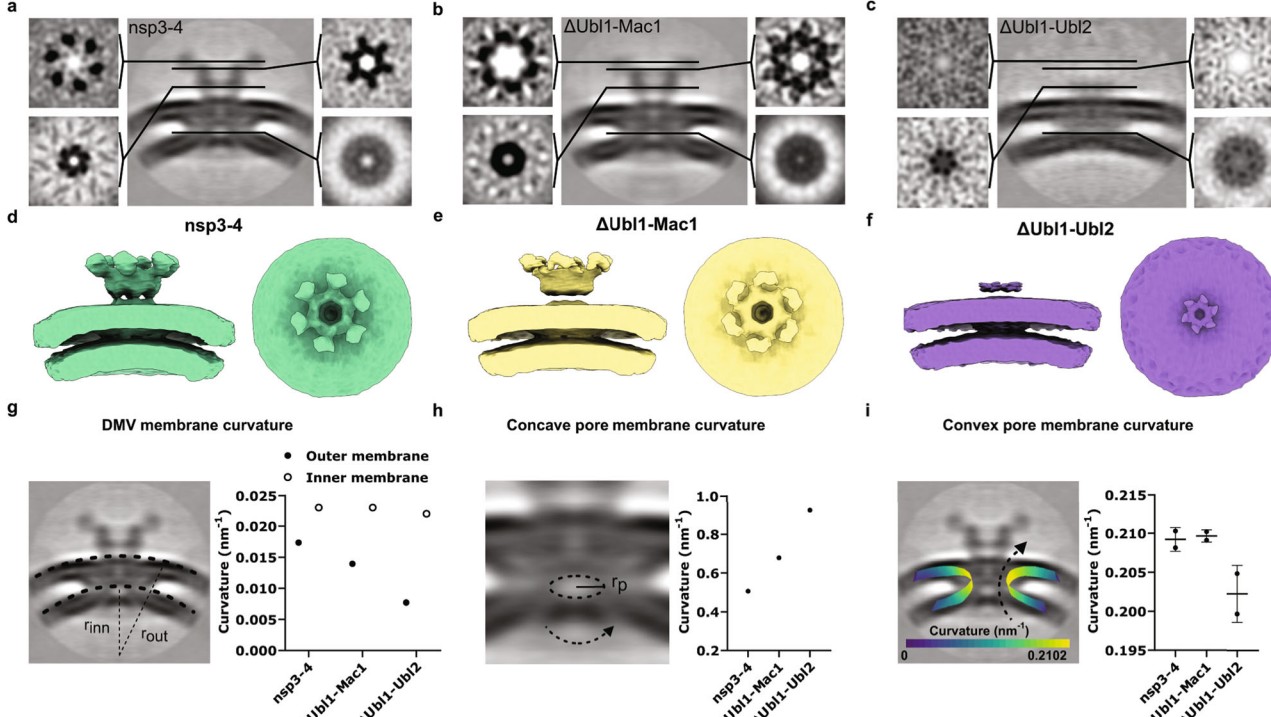

**Fig. 4 | The region spanning Ubl1-Ubl2 domains of SARS-CoV-2 nsp3 contributes to the structural integrity of the pore.** Orthogonal slices of filtered C6-symmetrized subtomogram averages obtained from tomograms of VeroE6 cells transfected with nsp3-4 (**a**), ΔUbl1-Mac1 (**b**), ΔUbl1-Ubl2 (**c**). The position of orthogonal slices along the z-axis is indicated by black lines in the cross-sectional view of the average. **d**–**f** Side (left) and top views (right) of filtered subtomogram averages displayed as isosurfaces. **g** Membrane curvature of DMV outer membrane changes upon truncation of nsp3. Curvature was calculated from the radius of the outer and inner DMV membrane using the subtomogram average. **h** Concave pore membrane curvature is altered upon truncation of nsp3. Curvature was calculated from the radius (r values: nsp3-4: 2 nm, ΔUbl1-Mac1: 1.5 nm, ΔUbl1-Ubl2: 1.1 nm) of the pore between the outer and inner membrane using the subtomogram average. **i** Convex pore membrane curvature is slightly altered in ΔUbl1-Ubl2. Curvature was calculated using a segmented subtomogram average. Data are presented as two measurements per subtomogram average with corresponding mean and SD.

and Mac3 which connect the prong with the central ring of the crown. The central ring mainly consists of DPUP, Ubl2 and PL^pro.

Since nsp4 includes predicted transmembrane domains between residues 10 and 385, its localization is likely confined inside the membrane region of the pore and not to the crown. Assuming that the pore is composed of six copies of nsp3 and nsp4, as determined by the symmetry scan of the average without imposing any symmetry, the size of the SARS-CoV-2 pore can be estimated at 1.6 MDa. The oligomeric status of the pore agrees with AlphaFold Multimer predictions using different oligomeric states of nsp4, which showed that interaction between subunits is predicted with higher confidence for pentameric and hexameric complexes (Supplementary Fig. 9).

To shed light on putative interactions between nsp3 and nsp4 within the membrane region of the pore we performed molecular dynamics simulations of predicted nsp3 and nsp4 dimer structures and examined their conservation during the COVID-19 pandemic (Supplementary Fig. 10). Structure prediction showed that the complex of nsp3 and nsp4 is mediated by interactions including that between residues nsp3 D1478, nsp3 Y1483, and nsp4 K67 localized in the luminal region of the DMV membranes (Supplementary Fig. 10d). In addition, these interactions were stable for 1 μs of all-atom molecular dynamics, with less optimal hydrogen-bonding geometry between nsp3 D1478 and Y1483 when nsp4 mutations H120N/F121L are added that were found to prevent SARS-CoV-1 replication[20] (Supplementary Fig. 10c–e).

## Discussion

The coronavirus DMV pore was discovered and structurally characterized in MHV infected cells. However, besides nsp3 the components of the MHV pore have not so far been identified and it has been proposed that other viral proteins such as nsp4 and nsp6 constitute the pore[1]. Although it has been shown that SARS-CoV-2 DMVs also contain a pore[1], its structure has not been determined, mainly due to the need for chemical fixation to inactivate the infected cells prior to cryo-FIB milling and cryo-ET. Here, we took advantage of a transfection-based system, which allows us to study SARS-CoV-2 pore minimal constituents at native conditions without the need of chemical fixation prior cryo-FIB milling and cryo-ET. Notably, we revealed that in the absence of SARS-CoV-2 other viral components, nsp3 and nsp4 are sufficient to form a pore that spans DMVs (Fig. 1). Interestingly, nsp3-4 induced DMV network is still clearly connected to the ER membrane showing that nsp6 is not crucial for linking the DMVs to the ER as suggested before[14]. In contrast to the DMVs found in infected cells, DMVs induced by nsp3-4 are uniform in size and do not show vesicle packets or convoluted membranes that are also known to contain nsp3[4,6]. Hence, the formation of vesicle packets and convoluted membranes may be driven by other viral proteins. Interestingly, we detect pores not only in the DMVs but also in the DMCs that together form a network of interconnecting DMVs. The observed pores inserted in DMCs show similarly oriented crowns on only one side of the double membrane. While the nsp3-4 pore has a similar shape and features as the pore found in MHV-infected cells[1], the structure of the prongs is different in the SARS-CoV-2 nsp3-4 transfected cells compared to the prongs found in MHV infected cells (Supplementary Fig. 8d). These differences between the prongs of SARS-CoV-2 nsp3-4 and MHV are likely reflecting the difference in the arrangement of the domains in the nsp3 N-terminal region. MHV nsp3 does not contain the Mac2 and Mac3 domains and contains a PL1^pro domain proximal to Mac1[21]. In addition, SARS-CoV-2 and MHV nsp3

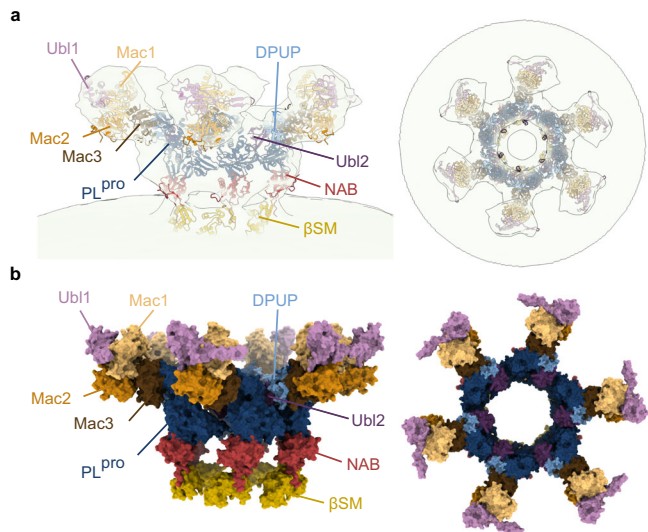

**Fig. 5 | Model of nsp3 domain localization in the crown region of the DMV pore.** **a** Side and top view of model showing X-ray-based crystal structures, nuclear magnetic resonance (NMR)-based structures and structures predicted with ColabFold fitted into the subtomogram average of the crown region. Ubl1 (7KAG), Mac1 (ColabFold), Mac2 (ColabFold), Mac3 (7XC3), DPUP (7P2O), Ubl2 (ColabFold), PL[pro] (7TZJ), NAB (ColabFold), βSM (7T9W). **b** Side and top view of model showing surfaces of nsp3 domains localized in the crown region.

proteins have 74 % amino acid similarity. Thus, our data indicate that pores induced by betacoronaviruses have a similar architecture but the crown region might be structurally more variable. Importantly, we cannot exclude that the observed differences between SARS-CoV-2 nsp3-4 and MHV pore are not due to another viral protein that is lacking in our minimal system or host cell proteins which would interact with a viral protein. Hence, a comparison of the SARS-CoV-2 pore with the SARS-CoV-2 nsp3-4 pore structure is needed to elucidate whether another protein is present in the pore in SARS-CoV-2 infected cells. This is, however, currently not possible due to the necessity to inactive SARS-CoV-2 infected cells by chemical fixation prior to cryo-EM studies, which has a negative impact on the overall structure of the crown[22].

In contrast to DMVs in MHV infected cells, overexpression of nsp3-4 leads to a slightly higher number of pores per DMV with decreased diameter. However, the number of pores per DMV does not correlate with the diameter of DMVs in our expression system (Supplementary Fig. 8g). From this data, we conclude that there is a defined number of pores per DMVs and that overexpression of nsp3-4 could lead to an increased number of pores. A higher number of pores could in turn be responsible for increased curvature and smaller DMVs (or vice versa) but the presence of RNA content has also been shown to influence the size of replication organelles in chikungunya virus (CHIKV) infected cells[23]. As shown by a cryo-ET and mathematical modeling, the membrane remodeling and the size of the replication vesicles (spherules) induced by CHIKV pores is controlled by the RNA content and the energy released by RNA polymerization[23]. It remains to be seen whether the presence of viral RNA and viral replication and transcription is responsible for the size difference between DMVs formed in SARS-CoV-2 infected and nsp3-4 transfected cells. Finally, attenuated SARS-CoV-2, which can be studied by BSL2, is needed to structurally compare the nsp3-4 and SARS-CoV-2 infection induced pores.

The truncation of Ubl1 and Mac1 showed collapsed prongs of the crown indicating that Ubl1 and Mac1 domains are important for localization and stabilization of Mac2 and Mac3 domains. Mac1 domain has adenosine diphosphate (ADP)-ribosyl hydrolase activity to counteract host-mediated antiviral ADP-ribosylation[24,25] and Ubl1 was shown to

interact with N protein[26–28] which argues for their localization at the tip of the crown to assure accessibility to interacting partners. Strikingly, deletion of the Mac2-Mac3-DPUP-Ubl2 domains leads to a crownless pore. The PL[pro] densities are not resolved in the subtomogram average after deletion of Ubl1-Ubl2 presumably because PL[pro] domains become flexible and do not anymore retain 6-fold symmetry. Overall, this is in agreement with a study using a mammalian two-hybrid system which revealed SARS-CoV-1 nsp3 self-interactions[29], and with a study showing that Mac and Ubl2 domains modulate the stability of the PL[pro] domain in porcine deltacoronavirus (pDCoV)[30]. Thus, Mac2-Mac3-DPUP-Ubl2 domains provide stability to the crown. Based on this data, peptides that are homologous to interaction interfaces in the Ubl1-Mac1-Mac2-Mac3-DPUP-Ubl2 domains or small molecules could be designed to interfere with the assembly of the crown ring and thereby with the function of the pore.

While uncleaved nsp3-4 is able to induce membrane pairing (Fig. 3d, h), it fails to form a pore. Hence it is plausible that the proteolytical cleavage allows rearrangement of the nsp3 and nsp4 into upside-down conformation which in turn locally bends the membrane required for pore formation. Since we can exclude the presence of nsp6 in our minimal system, the curvature of the membrane bilayer within the pore is likely driven by the rearrangement of the nsp3 and/ or nsp4 transmembrane domains upon cleavage. This is supported by our data showing that pore radius and both pore concave and convex curvature are altered in the absence of an intact crown (Fig. 4h, i). Therefore, we propose that the deletion of the Ubl1-Ubl2 region which leads to the loss of the crown also alters the interactions of nsp3 and nsp4 transmembrane domains lining the pore. Since our data show that deletion of Ubl1-Ubl2 from nsp3 leads to aberrant formation of DMVs (Fig. 3c), it is likely that pore assembly is required for DMV formation. A recent study showed the importance of reticulons in the nsp3-4 DMV biogenesis[31], which could directly interact with the nsp3-4 pore to control membrane pairing and curvature.

Interestingly, the residues nsp3 D1478, nsp3 Y1483 and nsp4 K67, which we predicted to mediate interactions between nsp3 and nsp4 luminal domains, have a very low tolerance to mutation during the COVID-19 pandemic[32], suggesting their importance in pore assembly. The remaining hydrophobic region consisting of nsp4 residues 280–398 is predicted to form a separate, helical transmembrane domain that we suggest spans the inner DMV membrane[33]. Furthermore, the mutation nsp4 T492I, which is present in all Omicron lineages, over 90% of Delta lineages known by September 2021, and the last surviving A lineage, A.2.5[34], is likely localized to the DMV interior. A recent study found that nsp4 T492I enhances viral replication via increased nsp4-nsp5 cleavage efficiency by the viral main protease, nsp5[35], and nsp4-nsp5 cleavage may be required for DMV pore formation. Since the nsp3-4 pore structurally resembles a nucleopore[36], we propose to term the proteinaceous pores induced by coronaviruses and other positive-strand RNA viruses "replicopores". Replicopores are associated with virus RNA replication and membrane remodeling, forming either DMV with multiple pores or spherules with a single pore. A recently published article showed that Flock House Virus (FHV) protein A is the minimal component of the FHV dodecamer replicopore formed on the mitochondrial outer membrane[37] and revealed a proto-crown at high resolution[38]. Similar replicopores were shown to be induced by alphaviruses nsP1 protein of CHIKV[3,39].

The methodology presented in this study can be applied to study the minial system of other replicopores. It allows circumventing the necessity of chemical inactivation applied for biosafety reasons and therefore excluding the possibility of nanoscale artefacts such as aggregation of membrane proteins caused by fixatives[22,40]. Finally, the established experimental framework opens an avenue for high-resolution determination of nsp3-4 replicopore, it will be instrumental to reconstitute viral genome replication by adding the RTC

components, and to investigate the role of other putative viral components or putative host cell factors which are involved in the DMV biogenesis.

## Methods

### Cell culture

HEK293T (293T ECACC, 1202201) cells were purchased from Sigma-Aldrich. VeroE6 cells were purchased from American Type Culture Collection (ATCC; Catalog #CRL-1586). Cells were maintained in Dulbecco's modified Eagle medium (DMEM, Thermo Fisher Science) with high concentration of glucose, GlutaMAX supplement, 100 U/ml penicillin, 100 µg/ml streptomycin, and 10% fetal bovine serum at 37 °C and 5% $CO_2$. For DNA transfection TransIT-LT1 Transfection Reagent was used according to the manufacturer's protocol (Mirus Bio LLC).

### Plasmids and antibodies

Codon-optimized synthetic DNA of 7416 bp expressing HA-nsp3-nsp4-V5 cDNA was ordered in 3 fragments from Biocat GmbH (Heidelberg) and assembled using overlap-extension PCR followed by insertion in pcDNA3.1 vector backbone using EcoRI-XbaI restriction sites. The expression of nsp3 and nsp4 from this plasmid and the induction of DMVs were confirmed previously[13]. Truncations were generated using In-Fusion HD Cloning Plus kit (TAKARA Bio). The amino acid sequences of the nsp3-4 constructs used in this study are listed in Supplementary Data 1. pN1-mApple plasmid (#54567) was purchased from Addgene. To detect the HA-tag and V5-tag by immunofluorescence microscopy and Western blot analysis, a rabbit anti-HA (Invitrogen, 71–5500), mouse anti-V5 antibodies (Santa Cruz Biotechnology, sc-271944) and mouse anti-GAPDH (Santa Cruz Biotechnology, sc-47724) were used as primary antibodies, respectively. The secondary antibodies Alexa Fluor 546 goat anti-rabbit (Invitrogen, A11010), Alexa Fluor 488 goat anti-mouse (Invitrogen, A11029) and Alexa Fluor 488 goat anti-rabbit (Invitrogen, A11034) were used for immunofluorescence microscopy. The secondary antibodies mouse anti-rabbit IgG-HRP (Santa Cruz Biotechnology, sc-2357) and anti-mouse IgG BP-HRP (Santa Cruz Biotechnology, sc-516102) were used for Western blot analysis.

### Western blot analysis

VeroE6 cells were seeded into 6-well plates at a seeding density of $0.4 \times 10^6$ cells/well. Cells were transfected with 2 µg DNA/well. At 24 h post transfection (hpt), cells were washed with cold PBS and lysed using 1 ml prechilled lysis buffer (2% SDS, 50 mM Tris-HCl, pH 7.4) supplemented with protease inhibitor cocktail in PBS (Roche) for 15 min. The cell lysate was centrifuged at $16,000 \times g$, 15 min, 4 °C. The supernatant was collected and mixed with Laemmli buffer (BioRad) and DTT (final concentration 50 mM). All samples were boiled at 95 °C for 10 min and separated by electrophoresis on SDS-polyacrylamide gradient gels (4–15%). Proteins were transferred to polyvinylidene fluoride (PVDF) membranes (Bio-Rad) using a Trans-Blot Turbo transfer system (Bio-Rad). Membranes were blocked with 5% milk in TBS-T (1 × TBS with 0.1% Tween-20) for 1 h at room temperature (RT). The PVDF membrane was washed for 3-times10 min with TBS-T and incubated in primary antibody solutions prepared by diluting 1:200 (anti-HA) or 1:1000 (anti-V5, anti-GAPDH) in TBS-T with 5% milk over night at 4 °C. Subsequently, membranes were washed 3 times with TBS-T and incubated in secondary antibody solutions prepared by diluting 1:1000 in TBS-T with 5% milk for 1 h at RT. Blots were washed 3 × 10 min with TBS-T and incubated in enhanced chemiluminescence substrate (Clarity Western ECL substrate, Bio-Rad) solution for 5 min at RT in dark. Images were acquired with Azure 400 Imaging System (Azure Biosystems).

### Fluorescence microscopy

VeroE6 cells were seeded into 12-well plates on coverslips at a seeding density of $0.12 \times 10^6$ cells/well. Cells were transfected with 750 ng DNA/well. Cells were washed with PBS at 24 hpt and fixed by 4% formaldehyde (PFA) diluted in PBS (16% PFA (E15710), Science Services) for 15–30 min at RT. Cells were washed twice with PBS and permeabilized with permeabilization buffer (0.5% Triton X-100 in PBS) for 5 min at RT. Cells were washed 3 × 5 min with PBS and blocked with blocking buffer (3% lipid-free BSA in PBS-T) for 1 h at RT. Cells were washed 3 times with PBS and incubated with primary antibody solutions prepared by diluting 1:200 in dilution buffer (1% lipid-free BSA in PBS-T) for 1 h at RT. Cells were washed 3 × 5 min with PBS and incubated in secondary antibody solutions prepared by diluting 1:500 in dilution buffer for 1 h at RT in dark. Cells were washed 3 × 5 min with PBS and incubated in DAPI mix (1:1000 dilution in PBS) for 1 min at RT. Cells were washed 2 × 5 min with PBS and once with deionized water and mounted on clean glass slides using a 7 µl mounting medium (Prolong Glass, Life Technologies). After drying, samples were analyzed with Leica TCS SP8 confocal laser scanning microscope, equipped with a 63× objective (numerical aperture 1.40; 1 Airy unit) and a Leica HyD hybrid detector. One slice was chosen to show in the images, and the brightness and contrast were adjusted to the same scale in ImageJ/FIJI[41]. Cells for transfection/co-transfection efficiency experiments were seeded as described above. Each well was transfected with 750 ng DNA/well (500 ng of HA-nsp3-4-V5 construct and 250 ng of mApple construct). The staining protocol was performed as mentioned above with the rabbit anti-HA primary antibody and Alexa Fluor 488 goat anti-rabbit secondary antibody. Samples were analyzed with Celldiscoverer 7 (Zeiss) (5x objective with double magnification).

### High pressure freezing and freeze substitution

Sapphire discs with 3.0 mm diameter and 50 µm thickness (Wohlwend GmbH) were cleaned with 100% ethanol and coated with a 15 nm carbon layer using a sputter coater (EM ACE600, Leica). An "F" letter was scratched on the carbon side to distinguish between the two sides of the disc, followed by overnight incubation of the discs at 120 °C. Prior to use, the carbon-coated discs were plasma-cleaned for 10 s (s) in a Solarus 950 (Gatan), sterilized in 70% ethanol, washed once with DMEM, and placed in 1 ml DMEM in a 35 mm dish with the "F" side facing up. The DMEM was removed and HEK293T cells were seeded on the sapphire discs ($0.18 \times 10^6$ cells in 2 ml DMEM per dish). Cells were transfected with 1 µg DNA/dish (co-tranfection of pN1-mApple plasmid). At 24 hpt, sapphire discs with transfected cells were assembled with 1-hexadecene coated specimen carrier Type A and B (Wohlwend GmbH) with cells facing the 100 µm deep cavity of carrier Type A. Cells were vitrified by high-pressure freezing at approximately 2,200 bar maintained for 370 ms with a cooling rate of 20,000 K/s using a Leica EM ICE. Sapphire discs with vitrified cells were transferred from liquid nitrogen to a freeze-substitution (FS) solution (0.1% uranyl acetate in anhydrous acetone) cooled to −90 °C and processed in automated FS system (EM AFS2, Leica). After washing with acetone, samples were infiltrated with Lowicryl HM20 and polymerized using UV light. The FS protocol was performed according to Supplementary Table 1.

### Ultramicrotomy and electron microscopy of resin sections

Lowicryl-embedded samples were sectioned using diamond knives (DiATOME) and a UC7 ultramicrotome (Leica). Sections with 200 nm nominal thickness were placed on 2 × 1 mm copper slot grids (Gilder) coated with support film (1% formvar or pioloform). Grids were imaged with a Talos L120C TEM operated at 120 keV and equipped with a Ceta-M camera with a 4k × 4k CMOS (Thermo Fisher Scientific). The whole grid was mapped at a magnification of 155 × and images were acquired at magnifications of 5,300×; 11,000× and 45,000× (corresponding pixel sizes at the specimen level: 26.44 Å, 13.35 Å and 3.28 Å respectively) using SerialEM[42].

## Plunge-freezing

To prepare samples for plunge-freezing, 35 mm cell culture dishes coated with a thin layer of polydimethylsiloxane (PDMS) were used to culture cells. Holey gold grids (200 mesh Quantifoil™ Au R2/2 grids) were placed into a PDMS coated dish and plasma-cleaned for 10 s in a Gatan Solarus 950 (Gatan). The PDMS-coated dish with grids was sterilized with 70% ethanol and washed twice with DMEM. The DMEM was removed and VeroE6 cells were seeded on the grids at a seeding density of $0.12 \times 10^6$ cells/dish. Cells were transfected with 1 μg DNA/dish (925 ng of nsp3-4 constructs, 75 ng of mApple construct). At 18–24 hpt, cells were stained with 1 μg/ml Hoechst (B2261) for 5 min and washed once with DMEM. Cells were plunge-frozen into liquid ethane using a Leica EM GP2 automatic plunge-freezer. The ethane temperature was set to −183 °C and the chamber to 25 °C and 80% humidity. An additional 2 μl medium was added to the grid just before plunge-freezing. Grids were blotted from the back with Whatman® Type 1 paper for 3 s. Grids were clipped into FIB-AutoGrids™ (Thermo Fisher Scientific) designed for FIB milling.

## Cryo-light microscopy and cryo-focused ion beam milling

Cryo-light microscopy was performed at −190 °C using cryo-CLEM wide-field microscope (Leica Microsystems) equipped with a 50x objective with a numerical aperture of 0.9. A map was acquired as a Z-stack (30 μm, 300 nm spacing) in a bright field channel, blue fluorescence channel (excitation wavelength: 325–375 nm, emission wavelength: 435–490 nm) and red fluorescence channel (excitation wavelength: 540–580 nm, emission wavelength: 592–668 nm) covering 1.2 × 1.2 mm area using LAS X Navigator software (Leica). The final map was stitched using the Cryo-CLEM/Stitch TileScan plugin available at https://github.com/Chlanda-Lab/cryoCLEM[43] in ImageJ/FIJI.

Cryo-focused ion beam milling was performed using Aquilos dual-beam cryo-focused ion beam-scanning electron microscope (cryo-FIB-SEM) (Thermo Fisher Scientific) with a cryo-stage cooled to −180 °C. Grids were mapped by cryo-scanning electron microscopy (cryo-SEM) and the cryo-LM map of the grid was correlated to the cryo-SEM map using the MAPS Software (Thermo Fisher Scientific). Transfected cells were recognized by the mApple fluorescence signal and selected for milling. After the application of the protective organo-metallic platinum layer cells were milled gradually in 5 steps with a stage angle between 15° and 18° using a gallium ion beam. The first four steps were carried out automatically using a modified Autolamella script[44] available at https://github.com/DeMarcoLab/autolamella. The last two milling steps were performed manually with a nominal thickness of 150 nm. Micro-expansion joints were used to minimize lamella bending[45].

## Cryo-electron tomography and tomogram reconstruction

Cryo-electron tomography was done using a Krios cryo-TEM (Thermo Fisher Scientific) operated at 300 keV and equipped with a post-column BioQuantum Gatan Imaging energy filter (Gatan) and K3 direct electron detector (Gatan) with an energy slit set to 15 eV. As a first step, lamellae were mapped at 8,700× (pixel spacing of 10.64 Å) using a defocus of −65 μm in SerialEM[42] to localize double membrane vesicles. Tilt series were acquired using a dose-symmetric tilting scheme[46] with the zero-angle set to 8° and a nominal tilt range of 68° to −52° with 3° increments with SerialEM. Records were acquired as movies at target focus ranging from −4 to −2.5 μm, electron dose per record of 3 e⁻/Å² and a magnification of 42,000× (pixel spacing of 2.156 Å). Beam-induced sample motion and drift were corrected using MotionCor2[47]. Tilt series was aligned using patch tracking or fiducials (2–5 nm large metal clusters of Pt and Au that are deposited on lamella during milling) and tomograms were reconstructed using R-weighted back projection algorithm using 3DCTF, dose-weighting filter and SIRT-like filter 10 in the IMOD software package[48]. Tomograms were denoised using Content Aware Image restoration (Cryo-CARE)[49]. Figures were prepared by averaging 3 or 5 slices of a tomogram.

## Subtomogram averaging and tomogram rendering

Pore densities were identified and extracted using a dipole model in Dynamo version 1.1.514[50] using a box size of 256 pixels and an initial template model was created by averaging approximately 30 pores using the orientations inferred by the dipole model. To create the first average of the nsp3-4 sample, pores were picked using a general box model in Dynamo and aligned against the initial template using a spherical mask without imposing any symmetry. To create the average of the ΔUbl1-Mac1 and ΔUbl1-Ubl2 sample, a cylindrical mask was used. For averaging of the ΔUbl1-Ubl2 sample, pores were picked manually with the dipole-oriented model to determine the orientation of the pores. The conical, azimuthal, translational search and angular increments were gradually decreased within 6 iterations using parameters for refinement of the average No.1 (Supplemetary Table S2). A symmetry scan was performed on the final average No.1 in Dynamo. Subsequently subtomogram averaging was performed using C6 symmetry where average No.1 was used as a new template to obtain an average No.2 using the same search parameters that were used for creating average No.1. The attained resolution was estimated using Fourier shell correlation with 0.5 and 0.143 criterion using a derived subtomogram averaging project with 2 references and from odd and even half-sets of particles. The C6-symmetrized map of average No.2 was visualized as isosurface in ChimeraX[51]. The isosurface in Fig. 1 was low-pass filtered to 20 Å whereas the isosurfaces in Fig. 4 and Supplementary Fig. 8 were low-pass filtered to 40 Å. The number of particles used for each nsp3-4 construct as well as the estimated resolution of all subtomogram averages is shown in Supplementary Table 3. Volume rendering was performed manually in Amira (Thermo Fisher Scientific) after tomogram denoising using Cryo-CARE. Subtomogram average of the nsp3-4 pore was placed into the rendering based on coordinates determined from Dynamo cropping table using ArtiaX toolbox[52] in ChimeraX.

## Measurements and statistical analysis

**Measurements of radius and curvature analysis.** In IMOD, the osculating circle (as contour model) was fitted at the outer DMV membrane and the radius (r) of the osculating circle was used as a measurement for the radius of the DMV. The curvature of the outer DMV membrane was calculated as $1/r$ ($\kappa$) in the tomograms. The same measurement was done for the inner and outer DMV membrane in the subtomogram average. DMV diameter for Supplementary Fig. 3 and 8g was measured manually in IMOD using the longest inner diameter. Pore diameter was measured in ImageJ/FIJI using line density profile. Curve fitting and calculation of diameter was done in MATLAB. Radius values were used to calculate concave pore membrane curvature. Subtomogram averages were segmented in IMOD and convex pore membrane curvature was calculated using the imodcurvature (kappa) command. Curvature values were stored as inverse of the radius in the model. Napari was used to adjust the colormap.

**Measurement of luminal spacing.** The luminal space of DMVs was measured as the distance between the outer leaflet of the outer membrane and the inner leaflet of the inner membrane. Line density profiles (~ 30 pixels in width, 0.41 Å/pixel) of ~30 nm in length were determined across the respective membrane using the plot profile tool in ImageJ/FIJI. The distances between the first and the fourth global maxima corresponding to the luminal space were then measured for each plot profile. For each DMV, three measurements were performed and the mean of them was reported.

**Measurement of the nearest distance between pores and number of pores per DMV.** The number of pores per DMV was counted manually in IMOD by creating models. Each project represented one tomogram, each contour represented a DMV, and each point represented a pore. The coordinates of each point were exported as.NFF files. The distance between each point and its nearest neighbor on the

same contour was calculated by the script[53] available at Zenodo. Statistical analysis was done by using unpaired two-tailed t-test.

## Structure prediction and molecular dynamics

The nsp3 region predicted by DeepTMHMM[54] to contain transmembrane helices was extended by 20 residues (nsp3 1391–1589), combined with nsp4 residues 1–124, and subjected to structure prediction using ColabFold[19]. Both AlphaFold2[55] and AlphaFold-Multimer[56] were used for prediction, both without templates, with 12 recycles, and using model 5. The logarithm of the mutation tolerance normalized by nucleotide-level mutation bias[32] was averaged over the first two nucleotides for each codon and compared to the distribution across ORF1a. Molecular dynamics started from the 200-ns timepoint of a previously reported nsp3-nsp4 simulation[33], building systems identically in a DOPE/POPC/SAPI bilayer, making point mutations with CHARMM-GUI Membrane Builder[57], and using previously reported simulation conditions. Simulated systems consisted of nsp3 residues 1404–1490 and nsp4 residues 1–124 with ACE and CT3 capping of truncated termini. Systems with wild-type and H120N/F121L nsp4 included 64,130 and 64,120 atoms, respectively, and included three predicted disulfide bonds. All histidines were uncharged. Trajectories were aligned to nsp4 residues 36–124 for visualization. Atomic coordinates were stored at 100-ps intervals and trajectories were analyzed using VMD[58]. Mac1, Mac2, Ubl2 and NAB domains were predicted with ColabFold using AlphaFold2, no templates and 3 recycle steps. Predictions showed predicted local distance difference test (pLDDT) values of ≥80 (Mac1), ≥50 (Mac2), ≥80 (Ubl2) and ≥50 (NAB). Models with the highest average pLDDT out of those predicted by the 5 AlphaFold2 models were used. Different models were used to test whether structures of nsp4 oligomers could be inferred from nsp4 sequence alignments using ColabFold. We first performed a RosettaFold2[59] prediction of full-length nsp4 with C6 symmetry. This was followed by predictions of nsp4 residues 31–397 without symmetry constraints, excluding nsp4 regions lacking predicted interactions in preliminary tests in order to not exceed available GPU memory. Five structures were predicted for each oligomer using AlphaFold Multimer with different weights and three recycles. The top scoring prediction was analyzed for each oligomer.

## Reporting summary

Further information on research design is available in the Nature Portfolio Reporting Summary linked to this article.

## Data availability

Electron tomography data have been deposited to the Electron Microscopy Data Bank under accession codes EMD-15925 (tomogram of nsp3-4 induced DMVs in Fig.1), EMD-15926 (tomogram of ΔUbl1-Ubl2 induced pores in Fig.3), EMD-15927 (tomogram of ΔUbl1-Mac1 induced pores in Fig.3), EMD-15928 (tomogram of GG > AA induced pores in Fig.3), EMD-15929 (tomogram of nsp3-4 induced pores in Fig.3), EMD-15963 (subtomogram average of nsp3-4 pore in Fig.1), EMD-15964 (subtomogram average of ΔUbl1-Mac1 pore in Fig.4), EMD-15965 (subtomogram average of ΔUbl1-Ubl2 pore in Fig.4). Files for the MD trajectories are available at Zenodo (https://doi.org/10.5281/zenodo.10069883). Source data are provided with this paper.

## Code availability

The script[53] used to calculate the nearest pore distance from IMOD models is available at Zenodo.

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

## Acknowledgements

We thank the Infectious Diseases Imaging Platform (IDIP) at the Center for Integrative Infectious Disease Research Heidelberg, the cryo-EM network at the Heidelberg University (HD-cryoNET) and Heidelberg University Electron Microscopy Core Facility for support and assistance. The authors gratefully acknowledge the data storage service SDS@hd supported by the Ministry of Science, Research, and the Arts Baden-Württemberg (MWK), the German Research Foundation (DFG) through grant INST 35/1314-1 FUGG and INST 35/1503-1 FUGG. This work was supported by a research grant from the Chica and Heinz Schaller Foundation (Schaller Research Group Leader Program). Work of P.C. and R.B. is supported by the Deutsche Forschungsgemeinschaft (DFG, German Research Foundation) project no. 240245660–SFB1129 and project number 437060729 (PC and MWM). In addition, work of R.B. is supported by DFG project no. 272983813–TRR 179) and by the project "Virological and immunological determinants of COVID-19 pathogenesis – lessons to get prepared for future pandemics (KA1-Co-02 "COVIPA")", a grant from the Helmholtz Association's Initiative and Networking Fund. V.P. is supported by a European Molecular Biology Organization (EMBO) long-term fellowship (ALTF454-2020). LZ and JM are supported by CoVLP project of the Flagship Initiative Engineering Molecular Systems. ZH received support for this work from FCT - Fundação para a Ciência e a Tecnologia, I.P., through MOSTMICRO-ITQB R&D Unit (UIDB/04612/2020, 510 UIDP/04612/2020) and LS4FUTURE Associated Laboratory (LA/P/0087/2020) and from a joint research agreement with the Oki-nawa Institute of Science and Technology.

## Author contributions

Conceptualization: L.Z., X.Z., P.C.; Methodology: L.Z., X.Z., J.M., M.W.M., V.P., Z.H., P.C.; Investigation: L.Z., X.Z., J.M., P.C.; Visualization: L.Z., Z.H.; Funding acquisition: R.B., P.C.; Project administration: J.M., P.C.; Supervision: R.B., P.C.; Writing – original draft: L.Z., P.C.; Writing – review & editing: L.Z., V.P., Z.H., R.B., P.C.

## Competing interests

The authors declare no competing interests.
