## [Peer Review File · Nature Communications]

SARS-CoV-2 nsp3 and nsp4 are minimal constituents of a pore spanning replication organelleEditorial Note: This manuscript has been previously reviewed at another journal that is not operating a transparent peer review scheme. This document only contains reviewer comments and rebuttal letters for versions considered at *Nature Communications*.

REVIEWER COMMENTS

Reviewer #1 (Remarks to the Author):

The manuscript describes in detail for the first time the minimal components required for SARS-CoV-2 pore formation. This is a revised version of a previous manuscript I reviewed. In the current manuscript the authors have responded to my earlier comments and have addressed some of my previous concerns. Most importantly, they now provide convincing curvature of membrane analysis in their sub-tomogram averages.

Nevertheless, the current manuscript still needs major improvement. The references should be carefully revisited.

For example, Page 3: line 23. Reference 18 not correct.

While it is clear the authors targeted mApple positive cells to perform FIB milling, worryingly, the authors don't reveal or show how the selection of mApple makes it more likely to target cells overexpressing nsp3/nsp4. In this respect some cells will overexpress one (mApple) or the other (nsp3/4), or both. Page 3, the sentence starting by "our previous work showed co-transfection..." reports on the finding that used co-transfection of IAV proteins. Co-transfection is not same for all plasmids and generalizing IAV proteins with mApple/nsp3-nsp4 is not convincing and does not necessitate the presence of mApple ensures the over-expression of nsp3/nsp4 in the same cell. All in all, it seems possible that these cells are likely to have expression of nsp3/nsp4, however I am not convinced that this approach works and increases throughput: rather it is something that worked in the subset of cells that they milled and would have worked regardless of mApple being present or not. I am unconvinced that this is "guided" FIB milling to target a specific cell or protein.

How the authors accurately targeted DMV clusters in lamella is not described in the methods. How many lamellae did they generate? How many tomograms were collected? There is no mention of the existence of search maps, although there must be some. It would seem appropriate if the authors showed a TEM search map of a representative lamella which would help the reader understand how they targeted DMV clusters for cryo-ET (surely, the DMVs are clearly visible in the TEM search map).

Page 10, the sentence "This indicates that the uncleaved nsp3-4 induces membrane pairing, but pore formation is required for membrane bending and scission which gives rise to DMVs..." is speculative and the significance of their findings is limited. In this respect the authors provide little evidence that the whorl structures are related to pore containing DMVs and/or contain uncleaved nsp3-nsp4. One possibility is, they could be autophagic structures derived from the ER unrelated to DMVs (this can be tested by colocalising with autophagosome markers). The authors also need to perform immuno-gold labelling to provide evidence these whorls contain uncleaved nsp3-4.

The data suggests other protein densities, such as obvious ribosomes (movies 1 and 2) are present in the DMVs (other than granules). It should be clarified in the paper that these structures are noticeable and captured in their data.

Reviewer #3 (Remarks to the Author):

The authors have satisfactorily addressed the comments raised in my previous review.

We would like to thank the reviewers for reading the revised version of the manuscript and for the additional comments and edits. We addressed the comments, formatted all figures with a new width and shortened the abstract to meet the word limit. The Word document is provided with tracked changes.

Reviewer #1 (Remarks to the Author):

The manuscript describes in detail for the first time the minimal components required for SARS-CoV-2 pore formation. This is a revised version of a previous manuscript I reviewed. In the current manuscript the authors have responded to my earlier comments and have addressed some of my previous concerns. Most importantly, they now provide convincing curvature of membrane analysis in their sub-tomogram averages.

Nevertheless, the current manuscript still needs major improvement. The references should be carefully revisited.

For example, Page 3: line 23. Reference 18 not correct.

Thank you for noticing this, we have corrected the reference accordingly.

While it is clear the authors targeted mApple positive cells to perform FIB milling, worryingly, the authors don't reveal or show how the selection of mApple makes it more likely to target cells overexpressing nsp3/nsp4. In this respect some cells will overexpress one (mApple) or the other (nsp3/4), or both. Page 3, the sentence starting by "our previous work showed co-transfection..." reports on the finding that used co-transfection of IAV proteins. Co-transfection is not same for all plasmids and generalizing IAV proteins with mApple/nsp3-nsp4 is not convincing and does not necessitate the presence of mApple ensures the over-expression of nsp3/nsp4 in the same cell. All in all, it seems possible that these cells are likely to have expression of nsp3/nsp4, however I am not convinced that this approach works and increases throughput: rather it is something that worked in the subset of cells that they milled and would have worked regardless of mApple being present or not. I am unconvinced that this is "guided" FIB milling to target a specific cell or protein.

This CLEM method allows us to guide milling but not to precisely target the nsp3-4 organelles, which was not claimed in the manuscript. The reason for that was that the transfection efficiency in Vero cells was rather low. This is not meant to target specific areas of the cell. To clarify this, we have replaced targeted with "guided" in the text and Supplementary Fig. 1.

The sentence on Page 3, Line 15-18 was changed accordingly: "This allowed us to perform cryo-correlative light and scanning electron microscopy to selectively cryo-FIB mill transfected cells expressing nsp3-4 without tagging the nsp3-4 with a fluorescent protein (Supplementary Fig. 1).

We used the mApple co-expression with nsp3-4 to select transfected cells for cryo-FIB milling. As correctly noted, this approach can increase the success rate of milling transfected cells only if the co-transfection efficiency of nsp3-4 and mApple is high. We now include new data to show that our transfection efficiency in VeroE6 cells is with 5.4 % rather low (analysis

of about 3000 cells). Our co-transfection efficiency of nsp3-4 and mApple is 80.6 % (Supplementary Fig. 1h-j) and hence, this approach dramatically improves the success rate of milling transfected cells with nsp3-4. One should note that the transfection efficiency is usually not the same in different fields of view and varies a lot within one batch of prepared grids which were prepared in the same 35 mm dish.

How the authors accurately targeted DMV clusters in lamella is not described in the methods. How many lamellae did they generate? How many tomograms were collected? There is no mention of the existence of search maps, although there must be some. It would seem appropriate if the authors showed a TEM search map of a representative lamella which would help the reader understand how they targeted DMV clusters for cryo-ET (surely, the DMVs are clearly visible in the TEM search map).

The number of tomograms is presented in Supplementary Table 3. An exemplary cluster of DMVs was provided in Supplementary Fig 1f. Since this was too small, we now provide an exemplary medium magnification map which shows a cluster of the nsp3-4 vesicles which is included in Supplementary Fig 1g. We have extended Supplementary Table 3 where we already reported the number of tomograms to also include the number of lamellae.

Page 10, the sentence “This indicates that the uncleaved nsp3-4 induces membrane pairing, but pore formation is required for membrane bending and scission which gives rise to DMVs...” is speculative and the significance of their findings is limited. In this respect the authors provide little evidence that the whorl structures are related to pore containing DMVs and/or contain uncleaved nsp3-nsp4. One possibility is, they could be autophagic structures derived from the ER unrelated to DMVs (this can be tested by colocalising with autophagosome markers). The authors also need to perform immuno-gold labelling to provide evidence these whorls contain uncleaved nsp3-4.

These structures were not visible in the untransfected control that we have already provided as part of the first revision. In addition, this structure was reported upon expression of MERS nsp3-4 GG→AA previously ([10.1128/mBio.01658-17](https://doi.org/10.1128/mBio.01658-17), Figure 5B), hence supporting that these are indeed induced by nsp3-4. We included a sentence to address this comment. See Page 7 line 2 (The whorl-like structures were not observed in untransfected HEK293T cells (Supplementary Fig 6)).

The data suggests other protein densities, such as obvious ribosomes (movies 1 and 2) are present in the DMVs (other than granules). It should be clarified in the paper that these structures are noticeable and captured in their data.

We now also mention that DMVs occasionally contain ribosomes. Page 4, Line 9.

Reviewer #3 (Remarks to the Author):

The authors have satisfactorily addressed the comments raised in my previous review.

Thank you for reading the manuscript again and for your previous comments.

REVIEWERS' COMMENTS

Reviewer #1 (Remarks to the Author):

The authors have addressed all my concerns and comments.